# Exploring the Structure of Human Adjective Representations

**Karan Grewal**
Department of Computer Science
University of Toronto
Toronto, ON, Canada
karang@cs.toronto.edu

**Joshua C. Peterson**
Department of Computer Science
Princeton University
Princeton, NJ, USA

**Bill Thompson**
Department of Computer Science
Princeton University
Princeton, NJ, USA

**Thomas L. Griffiths**
Department of Computer Science
Department of Psychology
Princeton University
Princeton, NJ, USA

## Abstract

Human semantic representations are both difficult to capture and hard to fully interpret. Similarity judgments of words are highly sensitive to context, and association norms may only capture coarse similarity. By contrast, feature norms are more interpretable, and the number of norms can be scaled without limit, but they often only exist for sets of nouns described with concrete norms. In this paper, we introduce a new large dataset of nouns normed by a set of continuous adjective ratings both concrete and abstract. We compare our dataset to other forms of representation and find that they capture rich, unique structure, which can be represented by a low-dimensional latent semantic space. We further make relationships between our data and neural network representations from different modalities. Our dataset contributes to an increasingly detailed picture of one relatively sizable swath of human semantic representations, and can be used in a variety of modeling paradigms.

## 1 Introduction

A fundamental component of human cognition is our capacity to represent complex concepts. Understanding the semantic representations that underpin people's ability to make judgments about familiar and novel concepts is a challenge that spans psychology, linguistics, and computer science. This challenge is difficult because people's semantic representations must be reverse-engineered through indirect methods. Moreover, the flexible, context-dependent nature of our semantic representations makes them difficult to characterize beyond specific contexts.

Empirical research designed to elicit association that under people's judgments have largely focused on similarity judgments between objects. In particular, pairwise similarity judgments and norms of association (De Deyne et al., 2019; Nelson et al., 2004) have been extensively studied by psychologists. However, this approach has two main drawbacks. First, the number of pairwise judgments grows quadratically with the number of concepts, and thus the cost and effort increases to collect them. Second, these judgments are heavily context-dependent. An alternative approach is feature production norms (McRae et al., 2005): this involves studies about specific properties of objects, such as whether an object is related to specific sensory modalities (Lynott et al., 2019), funny (Engelthaler & Hills,

3rd Workshop on Shared Visual Representations in Human and Machine Intelligence (SVRHM 2021) of the Neural Information Processing Systems (NeurIPS) conference, Virtual.

Table 1: A subset of adjectives and their corresponding concreteness ratings used in our study. The concreteness ratings were taken from Brysbaert et al. (2014).

| adjective | concreteness rating |
|-----------|---------------------|
| *liquid* | 4.72 |
| *blue* | 3.76 |
| *shiny* | 3.33 |
| *useful* | 2.14 |
| *scholarly* | 1.80 |

2018), or concrete (Brysbaert et al., 2014). These studies can result in more directly interpretable, less context-dependent representations of objects, and data collection is easier because it scales linearly in the number of objects. However, feature production norms are often collected independently, meaning that the objects that have been rated in one study are not necessarily the same objects that have been rated in another (Binder et al., 2016; Richie et al., 2019).

In this paper, we introduce a novel collection of objects and attributes from which both similarity can be inferred as well as normative judgments corresponding to specific properties. We make the following two contributions:

- We introduce a novel and rich dataset of noun-adjective applicability ratings which can be interpreted both as a) feature norms for adjectives over a large set of nouns, and b) feature norms for nouns over a set of adjectives that vary in their level of abstractness and encompass a wide range of descriptions,
- We explore the relationship between our data and both language- and vision-based neural network representations of common objects, just as Tuli et al. (2021) recently did between human vision, convolutional neural networks, and Transformers.

## 2 Dataset

In this section, we describe our dataset which is publicly available online.[1]

### 2.1 Nouns and Adjectives

We picked a set of 187 nouns and 128 adjectives for which to collect noun-adjective applicability ratings, i.e., human judgments. First, we decided on a set of 308 basic-level nouns that aren't too high-level nor too granular (for instance, *fruit* and *labrador* respectively). These 308 nouns comprise the different nouns we used in our experiments, and were taken from Wang & Cottrell (2016) who studied which ImageNet classes are subordinate, super-ordinate, or basic-level. However, considering that we also wanted to make quantitative comparisons against learned word representations (e.g., Word2Vec), we omitted all nouns specified by more than one word (e.g., *traffic light*) and this thus reduced our set to 187 nouns that can be described by a single word, such as *elephant*.

Second, we selected 128 common adjectives with the intention of covering a broad range of domains. Adjectives (e.g., *salty*) were taken from a large set outlined by Brysbaert et al. (2014), which also provides human ratings regarding how abstract or concrete an adjective is on a scale from 1 to 5. In addition to covering a large range of adjectives, we also aimed to have adjectives distributed uniformly across the continual concreteness scale. See Table 1 for an overview of the adjectives and their concreteness ratings. An exhaustive list of all nouns and adjectives can be found in Appendix A.

### 2.2 Data Collection

We recruited participants from Amazon's Mechanical Turk platform to rate the extent to which a noun can be described by a given adjective. Responses were provided on an integer scale ranging from 0 (not at all) to 10 (absolutely). In an attempt to ensure ratings within an adjective were consistent, participants rated 25 randomly-chosen noun stimuli simultaneously for a given adjective.

---

[1]OSF repository: `https://osf.io/n934t/`

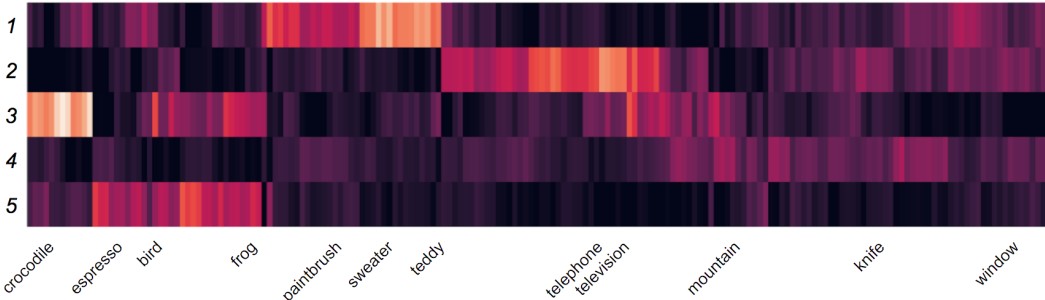

Figure 1: A visualization of the noun-adjective ratings matrix (transposed) after non-negative matrix factorization to reduce the adjective space from 128 dimensions to 5. Brighter colors indicate that a particular noun has a higher score along a certain dimension. We labeled just a few nouns.

This allowed participants to revisit ratings relative to each other. In a single survey, participants were asked to provide noun-adjective ratings for 4 adjectives, resulting in 100 ratings per survey. For each noun-adjective combination, at least 10 ratings were collected. As all noun-adjective combinations were included as part of the data collection process, we collected more than 187 nouns $\times$ 128 adjectives $\times$ minimum 10 ratings per pair $\approx$ 240,000 ratings in total. Participants were paid \$0.02 for each rating they provided, and were allowed to participate multiple times in the study since the noun-adjective they were assigned to rate were unlikely to reappear. See Appendix C for a visual illustration of the dataset.

## 3 Results

### 3.1 Intuitive Theories of Objects

What are the components of people's intuitive theories of objects? Over two thousand years ago, the Greek philosopher Empedocles put forward the theory that all objects in the universe are composed from a delicate balance of four basic components: earth, air, water, and fire. This theory may seem overly simplistic to modern readers, but the idea that people may represent objects in terms of a lower-dimensional semantic space is not simplistic at all. When people make semantic judgments, the representations that underpin these judgments can be thought of as potentially low-dimensional intuitive theories. The structure of these intuitive theories may reveal important generalizations about the features from which people's semantic representations are derived.

To examine this possibility, we tried to identify lower-dimensional structure in our dataset by applying non-negative matrix factorization to the adjective-by-noun matrix of human ratings (where each entry was the mean rating collected for a particular noun-adjective pair). Through this process, we identified five components which explain the structure in our dataset: the components that best explain people's judgments correspond to how (i) soft-textured, (ii) technology-based, (iii) animate, (iv) mobile, and (iv) edible an object is. Figure 1 visually shows how our set of nouns are represented across the five dimensions after dimensionality reduction. We also computed pairwise correlations between each row in Figure 1 and each row in the original adjective-by-noun matrix to get a sense of what the latent dimensions may correspond to, and these are listed in Table 2. Are the five adjective dimensions uncovered by non-negative matrix factorization orthogonal to each other, and can they thus be interpreted as a basis set of adjectives given our chosen nouns? It behooves us to raise this question, and we hope future work will elicit a response.

### 3.2 Predicting Human Ratings

Next, we explored how well existing representations of nouns capture knowledge about adjective applicability. That is, we set out to explore: given a noun and its representation, can we reliably predict the mean noun-adjective rating ascribed to a particular adjective by humans? We focused on representations taken from neural network models pertaining to linguistic and visual modalities. Our language-based representations were noun embeddings based on distributional semantics: Word2Vec

Table 2: The adjectives with the strongest correlations to each dimension uncovered by non-negative matrix factorization, according to our noun-adjective ratings. The representation for each adjective is simply a vector where each entry is the mean rating for its applicability to a particular noun.

| | |
|---|---|
| (top) row 1 | *cottony* (0.81), *soft* (0.80), *silky* (0.72), *woolen* (0.71), *pink* (0.69) |
| row 2 | *technological* (0.83), *electronic* (0.82), *metallic* (0.63), *organic* (0.62), *odorless* (0.57) |
| row 3 | *vertebrate* (0.74), *animate* (0.73), *intelligent* (0.72), *autonomous* (0.71), *brave* (0.69) |
| row 4 | *hard* (0.66), *immobile* (0.66), *wooden* (0.65), *concrete* (0.62), *mobile* (0.59) |
| (bottom) row 5 | *edible* (0.86), *organic* (0.82), *chewy* (0.76), *inorganic* (0.76), *tasteless* (0.73) |

(Mikolov et al., 2013) and GloVe (Pennington et al., 2014). To obtain visual features, we passed images of our nouns through trained convolutional neural networks and simply used the hidden activation values in the penultimate layer. To get a reliable estimate of the visual features for each noun, we passed the entire set of images found in ImageNet (Deng et al., 2009) corresponding to a particular noun through the network and then performed a final element-wise average. As vision researchers have investigated different architectures for image classification, we picked VGG-16 (Simonyan & Zisserman, 2015) and ResNeXt (Xie et al., 2017) since these are two reliable, well-cited methods.

Following a similar protocol to that of Richie et al. (2019), we used ridge regression to find a linear mapping from representations to human ratings. We learned a separate model for each adjective and for each type of representation (e.g., for the adjective *blue*, we separately fitted representations from each modality onto human ratings across all nouns). Each model was trained to minimize mean-squared error between its predicted and actual values, and we performed a grid search to select the ridge coefficient. We used leave-one-out cross validation, and computed a final $R^2$ value between all predicted and actual ratings.

As all 128 adjectives had previously been rated for concreteness (Brysbaert et al., 2014), we assessed the performance of each representation as a function of concreteness. Figure 2 illustrates our results when regressing to human ratings, and we make two remarks. First, it's intuitive to think that language better models abstract concepts such as *democracy* which the visual world has barely any way to represent. Indeed, we find that for relatively abstract adjectives, noun representations based on distributional semantics (i.e., language data) better predict human ratings than do visual features. This finding was statistically significant as we performed four paired sample $t$-tests (for each combination of language and vision representation) where we considered $R^2$ values predicted for the most abstract adjectives. When comparing the best language-based representation against the best vision-based one (i.e., GloVe vs ResNeXt), we obtained $t(7) = 3.53, p < 0.001$ and $t(15) = 6.98, p < 0.001$ taking into account the 16 and 32 most abstract adjectives, respectively.

Second, despite distributional semantics and visual features being completely unrelated in how they learn to represent concepts, it's interesting that they both predict human ratings with roughly the same accuracy ($R^2 \approx 0.3$) for the remaining adjectives that aren't too abstract. This naturally raises the question: despite differences in data modality and the objective function used for learning each type of representation, do distributional semantics and visual features ultimately encode the same information? Based on the results in this subsection, it certainly seems possible.

## 3.3 What Language Reads That Vision Doesn't See (And Vice-Versa)

Finally, following up to the results in Section 3.2, we set out to determine whether distributional semantics and visual features encode the same information about human judgments of noun-adjective applicability. For instance, is there something that Word2Vec/GloVe captures about *silky* things that visual features do not? To answer this, we measured the partial correlation between a) human ratings and GloVe embeddings while controlling for ResNeXt features, and b) human ratings and ResNeXt features while controlling for GloVe embeddings. We particularly chose GloVe and ResNeXt as they were the best performing language- and vision-based representations in our regression study, respectively.

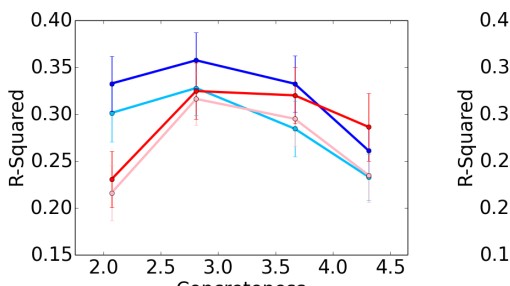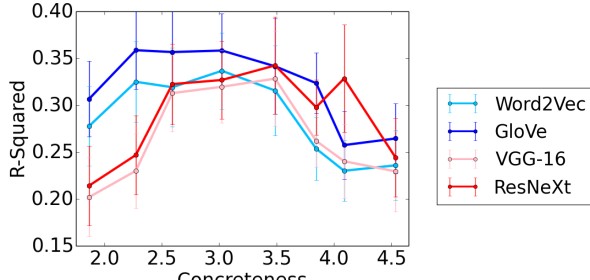

Figure 2: Average $R^2$ values per adjective bin as a function of concreteness when regressing noun representations to human ratings. Bins are simply adjective groupings based on concreteness to eliminate noisy results, and we show results for 4 (left) and 8 (right) contiguous adjective bins. We show standard error bars for each type of representation within each adjective bin.

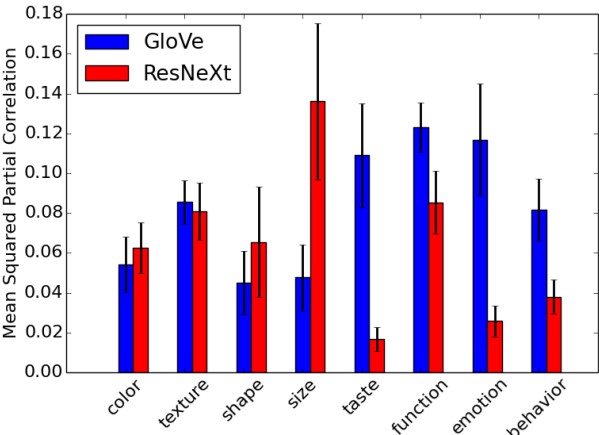

Figure 3: Mean squared partial correlation values between human ratings and GloVe/ResNeXt while controlling for the other. Adjectives were first grouped into one of eight domains (see Appendix B) before averaging was performed. We include standard error bars.

For this analysis, we divided adjectives into one of the following "domains": color, texture, shape, size, taste, function, emotion, and behavior.[2] See Appendix B for an exhaustive list of how each adjective was grouped. Each partial correlation score was retrieved per adjective for both GloVe and ResNeXt, and then we averaged the square of these values within each domain.[3]

Figure 3 reports the mean squared partial correlation within each domain. Not too surprisingly, GloVe explains much more unique variance in human ratings for adjectives describing taste, function, emotion, and behavior, which are largely non-visual domains while ResNeXt does the same for shape and size adjectives, which indeed have more of a visual component. For adjectives describing color and texture, no significant difference in amount of unique variance explained between either representation was found. We have now shown that distributional semantics and visual features encode different information about the nouns they represent, but why both types of representation achieve roughly the same $R^2$ value when predicting human ratings remains an open question.

---

[2]These domains are not well-defined. Instead, they are rather arbitrary and hand-picked, but nonetheless reasonable groupings.

[3]As the partial correlation values depend on a regression between variables, we followed the same protocol as described in Section 3.2 when regressing to human ratings.

# 4 Discussion

In this paper, we presented a large dataset of noun-adjective applicability ratings through crowd-sourcing human judgments which we also make publicly available. By analyzing the structure in our dataset, we identified a small number of organizing dimensions that capture people's intuitive theories of common objects and concepts. We then performed regression analyses to predict people's judgments from a variety of pre-existing semantic representations. We found that, for relatively abstracts adjectives, representations induced from distributional semantics tend to outperform representations induced from neural networks trained on image datasets. However, we observed evidence that purely visual features from deep neural networks may also explain unique variance in human ratings, reaffirming the notion that people's semantic representations integrate knowledge from the visual world as well as knowledge from language. Our results contribute to an increasingly detailed picture of the representations that underpin people's semantic knowledge, and how these representations relate to the knowledge acquired by contemporary machine learning algorithms.

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

# A Nouns and adjectives

In this section, we list all nouns and adjectives used in our study. First, here are the 187 nouns for which we collected ratings, listed alphabetically.

| Nouns (sorted alphabetically) |
| --- |
| abacus, acorn, aircraft, altar, ant, armor, artichoke, ATM, bag, bannister, basket, bear, bee, beetle, binder, bird, bookcase, bowl, box, bra, brake, bridge, broom, bubble, bus, butterfly, cabinet, camel, camera, cap, cassette, castle, cat, centipede, chain, chainsaw, cliff, cloak, closet, coil, comforter, compass, computer, cover, crab, crane, crayfish, crocodile, cucumber, curtain, dam, desk, diaper, dock, dog, doormat, dough, drill, elephant, envelope, eraser, espresso, fan, fence, fish, flower, flowerpot, fly, fountain, fox, frog, fungus, garbage, geyser, glass, gown, grille, guillotine, hat, hatchet, hay, heater, helmet, hippopotamus, insect, ipod, iron, jean, joystick, keyboard, knife, lamp, lampshade, lighter, lighthouse, lizard, llama, lobster, machine, mask, maze, memorial, menu, mitten, modem, mollusk, monitor, mop, mountain, mower, mushroom, muzzle, necklace, oscilloscope, packet, paddle, paintbrush, pajama, parachute, patio, pen, pencil, pepper, photocopier, pillow, plow, powder, printer, prison, puck, racket, radiator, radio, remote, rig, robe, roof, rug, salamander, sawmill, scoreboard, screen, seat, seatbelt, shaker, shield, ski, sled, snake, spoon, squash, stage, stethoscope, stick, store, stretcher, suit, sunglass, sweater, switch, syringe, t-shirt, table, teddy, telephone, telescope, television, tent, tie, toad, toilet, tray, triceratops, turtle, tusker, vacuum, valley, vehicle, vessel, walkingstick, wallet, whale, whistle, window, wing, wolf, worm |

Next, we give all 128 adjectives listed in decreasing order of concreteness rating according to Brysbaert et al. (2014).

| Adjectives (sorted by concreteness) | | | | |
| --- | --- | --- | --- | --- |
| human (4.93) | metallic (4.03) | feathery (3.62) | weak (2.79) | addictive (2.22) |
| rubber (4.86) | invertebrate (4.00) | opaque (3.62) | quiet (2.76) | unusual (2.21) |
| liquid (4.72) | sweet (4.00) | short (3.61) | odorless (2.72) | silly (2.20) |
| orange (4.66) | thick (4.00) | symmetrical (3.61) | old (2.72) | popular (2.16) |
| wooden (4.61) | pointy (3.97) | edible (3.55) | organic (2.68) | artistic (2.14) |
| concrete (4.59) | furry (3.96) | tall (3.36) | lactic (2.65) | professional (2.14) |
| blonde (4.52) | juicy (3.96) | curvy (3.33) | attractive (2.64) | stylish (2.14) |
| fat (4.52) | pink (3.93) | shiny (3.33) | flexible (2.64) | useful (2.14) |
| hairless (4.52) | salty (3.93) | fast (3.32) | sexy (2.64) | dangerous (2.13) |
| hairy (4.48) | round (3.90) | electronic (3.30) | sporty (2.62) | unfriendly (2.12) |
| wet (4.46) | unsalted (3.90) | slow (3.28) | cottony (2.61) | technological (2.08) |
| solid (4.42) | white (3.89) | tropical (3.25) | happy (2.56) | boring (2.07) |
| vertebrate (4.38) | soft (3.88) | audible (3.23) | tasteless (2.54) | autonomous (2.00) |
| frozen (4.34) | woolen (3.86) | ugly (3.23) | angry (2.53) | exciting (1.96) |
| hot (4.31) | cold (3.85) | asymmetrical (3.21) | childish (2.52) | adventurous (1.95) |
| yellow (4.30) | rough (3.83) | chewy (3.19) | funny (2.50) | usual (1.90) |
| dark (4.29) | thin (3.83) | transparent (3.18) | intelligent (2.46) | strange (1.86) |
| red (4.24) | dry (3.77) | strong (3.14) | nutritious (2.45) | unfamiliar (1.83) |
| dirty (4.23) | black (3.76) | expensive (3.13) | everyday (2.43) | scholarly (1.80) |
| automotive (4.19) | blue (3.76) | smelly (3.07) | inorganic (2.43) | traditional (1.76) |
| silky (4.12) | hard (3.76) | animate (3.04) | tasty (2.43) | godly (1.69) |
| grey (4.11) | loud (3.73) | scary (3.00) | feminine (2.41) | bad (1.68) |
| rectangular (4.11) | little (3.67) | mobile (2.93) | regular (2.40) | good (1.64) |
| green (4.07) | big (3.66) | immobile (2.86) | dumb (2.36) | normal (1.40) |
| triangular (4.07) | unsweetened (3.66) | unbreakable (2.82) | friendly (2.32) | |
| purple (4.04) | squishy (3.64) | new (2.81) | brave (2.26) | |

# B  Adjective domains

The table below shows how all 128 adjectives were grouped for the partial correlation analysis in Section 3.3.

| domain | adjectives |
|---|---|
| color | *black, blonde, blue, dark, green, grey, opaque, orange, pink, purple, red, transparent, while, yellow* |
| texture | *cold, concrete, cottony, dry, feathery, frozen, furry, hairless, hairy, hard, hot, juicy, liquid, metallic, pointy, rough, rubber, silky, squishy, soft, solid, wet, wooden, woolen* |
| shape | *asymmetrical, curvy, rectangular, symmetrical, triangular* |
| size | *big, fat, little, short, tall, thick, thin* |
| taste | *chewy, salty, sweet, tasteless, tasty, unsalted, unsweetened* |
| function | *addictive, animate, artistic, audible, autonomous, electronic, everyday, dangerous, fast, flexible, immobile, loud, quiet, mobile, slow, sporty, strong, stylish, technological, unbreakable, useful, weak* |
| emotion | *angry, boring, exciting, funny, happy, scary* |
| behavior | *brave, childish, dumb, friendly, intelligent, professional, scholarly, silly, unfriendly* |

# C  Dataset Sample

Below, we show a visual of our dataset. We present a subset of all nouns and adjectives, and the mean ratings for each pair.

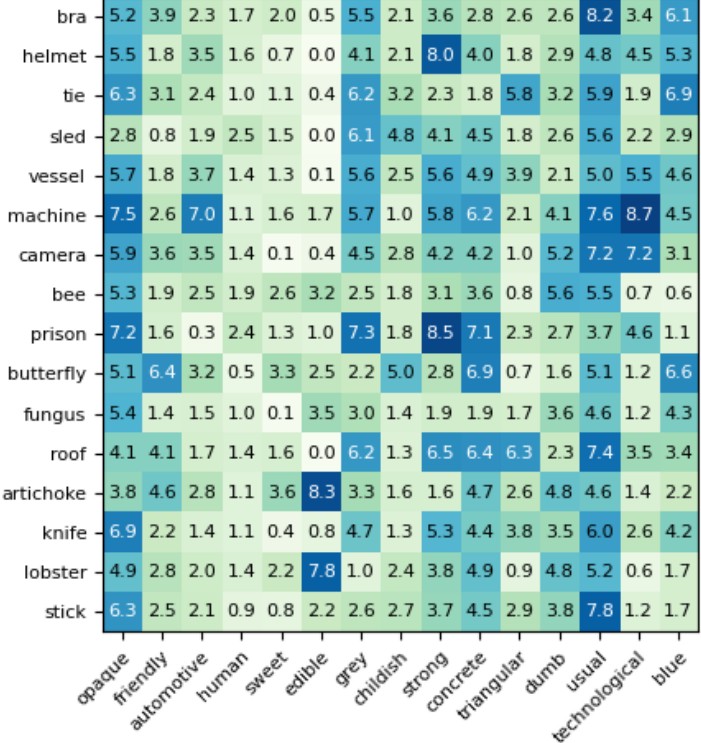

