# OpenReview forum: "Exploring the Structure of Human Adjective Representations"
_NeurIPS.cc/2021/Workshop/SVRHM — SVRHM 2021 Poster_

### Official Review · Reviewer_mTW6 · 2021-10-26
**Nice potential for dataset but missing validation and unclear motivation**

**Rating:** 4
**Confidence:** 5

**Review:**

The authors present a nice dataset of noun-adjective pairs: they collected a large dataset of continuous ratings for which adjectives match existing objects. In addition, they ran a number of interesting analyses exploring the structure of this dataset. I will center my review on these two aspects which were pointed out by the authors as the key contributions.

**Dataset.**
I think this dataset of feature norms might have the potential to make a good contribution to the scientific community. However, important methodological details are missing, which make it impossible to fully evaluate its usefulness. 10 ratings per pair are not a lot - specifically for online studies - and there are no numerical estimates of the internal consistency of ratings or the degree to which they align with existing norms. This, however, is crucial for the dataset to become useful for the community. In the context of the authors' predictions, the quality of the dataset becomes crucial. Without the Bayes accuracy, i.e. the best performance any model can achieve (which is given by the rating consistency, e.g. split-half correlation corrected with the Spearman-Brown formula), it becomes difficult to put model performance into context. Likewise, the selection of objects and adjectives seems a little ad hoc. Why limit the selection to the somewhat arbitrary set of concepts in ISLCVR? Shouldn't the neural network generalize to other concepts anyway (or even better since it might overfit to the concepts it had been trained on)? And what is the reasoning for the chosen set of 128 adjectives? I think the selection might be fine if the results generalize well to other datasets and show high consistency, but without this information, the value of the dataset remains unclear.

**Analyses.**
As I had mentioned, the analyses seem interesting but I am unsure about the motivation for them. What are we supposed to learn from the non-negative matrix factorization? Why were 5 dimensions chosen? Are the dimensions supposed to be interpretable, is there data to support this, and if they are interpretable, what does this tell us? Why do later analyses not rely on these factors but on external (again somewhat arbitrary - as acknowledged by the authors) groupings? Again, I see a lot of potential for these analyses but it remains somewhat unclear to me what they are supposed to tell us, and important methodological details are missing.

The comparison between the semantic embedding and deep neural network seem interesting, as well, but again, the motivation is unclear. Is the motivation that some types of adjectives may be better predicted by semantics and others by vision? If so, what would we learn that we didn't already know? And since the vision models are trained to categorize objects, to what degree can we truly distinguish between visual and non-visual features?

**Literature.**
In this context, there is a vast existing literature looking at object properties (e.g. featural vs. dimensional accounts) but the authors' discussion of previous existing work from cognitive science is somewhat superficial for a conference paper.

**Summary.**
Together, I think these data and analyses present a nice first step exploring the relationship between object nouns and adjectives but I think a lot more information (and more analyses) are needed to evaluate their usefulness, as well as a clearer motivation for what we learn from these analyses.

---

### Official Review · Reviewer_B1Ji · 2021-10-29
**Interesting work, potentially useful dataset, but analysis results seem unimpressive**

**Rating:** 6
**Confidence:** 3

**Review:**

The paper's contributions are mainly a new dataset consisting of human ratings of noun-adjective applicability, plus some structural analysis on top of it.

The dataset itself may be useful to the community, but I seem to have missed a statement about whether the authors will make it publicly available. It seems that would be a valuable contribution in its own right.

With regards to the analysis:

Section 3.1: It would be good to know how the authors arrived at the number five for the dimensionality reduction of the adjective space. Was this an ad-hoc choice, or did the authors search for a dimensionality that yielded meaningful results? Would one be able to derive meaning out of an NMF result with a different number of dimensions as well? How important is the choice of target dimensionality for this result?

Section 3.2: An R^2 value of 0.3 doesn't seem very impressive. I wonder if better prediction accuracy could have been achieved using nonlinear regression (i.e. training a classifier network for the cross-representation prediction task). Perhaps the assumption that the  human ratings can be decoded linearly from the representations is overly restrictive? In addition, while the difference between visual and linguistic representations for highly abstract adjectives may be logical and statistically significant, why does the R^2 value drop for both visual and linguistic representations for adjectives that are considered highly concrete? The authors don't make an attempt to explain this, as far as I can tell.

Overall, a paper with interesting ideas, but the results don't seem to be very convincing.

---

### Official Review · Reviewer_V1Km · 2021-10-30
**Concise, potential for significance, would be curious to see one more set of experiments.**

**Rating:** 7
**Confidence:** 5

**Review:**

Paper was clear and concise, easy to follow.

Line 25: Nit: effort effort. Increases?
Line 56: a many domains.

Suggest the authors try out CLIP text and image embeddings. CLIP has been trained explicitly with an objective to minimize the distance in embedding space, between language and image that are conceptually similar. This could also produce interesting results in remark #1: in the example given with the word "democracy", though abstract, there can be image associations when one hears the word, such as a ballot box or an ancient Athenian Agora or a statue of Cleisthenes. I'd be curious to see whether using language and image embeddings coming from a model like CLIP that has been trained with the aforementioned objective and on images and captions from the entire internet, would yield a smaller gap between how well such words are represented by distributional semantics versus visual features, or in general whether it would change results in a significant way. In a similar vein, with CLIP having produce SoTA resutls in mind as an example, for remark #2: I would argue that one could say CLIP gave a strong signal that distributional semantics and visual features ultimately encode the same information despite their differences in data modality. Not by directly learning through human similarity judgements but indirectly through making the assumption that captions and images on the internet are paired based on how humans saw fit. Thus, this observation per se is not novel.

Another downside is concerns about scalability of the proposed dataset. This could be a really cool dataset to be released as a large scale dataset, however scaling it sounds like it would require much hand crafting and engineering.

However, this work is particularly interesting because it studies the semantic power of language versus vision through the lens of parts-of-speech. Dividing words into adjectives and nouns and looking into how humans and neural networks perceive their correlation takes a closer and more targeted view on how unique or shared encoded information is between the two modalities. Also, if authors find a way to make their dataset scalable and include a wider variety of the language, that has potential to be a great contribution to the research community which is the significance of this work.

---

### Official Review · Reviewer_4oW7 · 2021-10-31
**Unsure that this necessarily makes much sense**

**Rating:** 5
**Confidence:** 1

**Review:**

I'll start off by making it clear that I'm not very confident in reviewing this paper, and I found bits rather hard to follow (not because of the writing, but because of a seemingly assumed knowledge of psycholinguistics). My understanding is that the authors wish to explore the relationship between vector representations (specifically word embeddings or visual embeddings) of nouns and the applicability of adjectives to those nouns as provided by human annotators.

The first set of experiments looks at whether given a noun embedding (either visual or word-based) it is possible to learn a linear relationship for the applicability rating for each adjective using ridge regression. Firstly I would ask why the authors choose to use ridge regression - why would one assume that certain elements of the embedding are relevant (or not) for this prediction task? I don't think this is deal-breaking, but it would be good to have some justification for the choice.  Secondly, one wonders if for the visual representations the averaging of representations might have some detrimental effect for cases where there is potential significant diversity in the images corresponding to the particular noun?

Overall, I guess I have a problem in understanding if the experiments show something meaningful/useful or are just showing artefacts of how the data was processed/analysed (as well as the inherent biases from the way the embedding models were trained). That being said, I do see some value in the dataset that has been constructed, as it could be used in the future when the aforementioned issues are addressed.

---

### Decision · Program_Chairs · 2021-11-02

Accept (Poster)